

# Impacts of spatial resolutions on projected changes in precipitation extremes: from site- to grid- scales

Jianfeng Li[1], Thian Yew Gan[2], Yongqin David Chen[3,4], Qiang Zhang[5,6,7], Zengyun Hu[8,1], Xihui Gu[9,1]

[1]Department of Geography, Hong Kong Baptist University, Hong Kong, China
[2]Department of Civil and Environmental Engineering, University of Alberta, Edmonton, Alberta, Canada, T6G 2W2
[3]Department of Geography and Resource Management, The Chinese University of Hong Kong, Hong Kong, China
[4]Institute of Environment, Energy and Sustainability, The Chinese University of Hong Kong, Hong Kong, China
[5]State Key Laboratory of Earth Surface Processes and Resource Ecology, Beijing Normal University, Beijing 100875, China
[6]Academy of Disaster Reduction and Emergency Management, Beijing Normal University, Beijing 100875, China
[7]State Key Laboratory of Earth Surface Processes and Resource Ecology, Beijing Normal University, Beijing 100875, China
[8]State Key Laboratory of Desert and Oasis Ecology, Xinjiang Institute of Ecology and Geography, Chinese Academy of Sciences, Urumqi 830011, China
[9]School of Environmental Studies, China University of Geosciences, Wuhan, China

*Correspondence to:* Jianfeng Li (jianfengli@hkbu.edu.hk)

**Abstract:** Precipitation extremes are localized and spatially heterogeneous events. Magnitude of precipitation extreme $p$ is expected to be spatial resolution dependent. Heavy precipitation extremes tend to be less intensive at coarser resolutions due to the averaging effect of the neighbouring less extreme events. Given the resolution dependent $p$ , this study aims to investigate how spatial resolutions affect projected changes in precipitation extremes between future and historical periods,
i.e. $p_{fut} - p_{his}$, which is a commonly used metric in climate projections. Our results show that although $p$ is sensitive to spatial resolutions, differences in $p_{fut} - p_{his}$ among various spatial resolutions are relatively small. Assessments of performances of GCMs in simulating $p$ and $p_{fut} - p_{his}$ are conducted based on three commonly used strategies that account for differences in spatial resolutions between GCMs and observations, i.e. the site-scale, the grid-scale, and the grid-point (i.e. direct comparison of grid- against scale- extremes) comparisons. Performances of GCMs in the site-scale comparison
outperform those in the grid-scale and grid-point comparisons, because the statistical downscaling method incorporates the site-scale information to the future values when downscaling the GCMs. Assessment results of the grid-point comparison are comparable to those of the grid-scale comparison, even though the former has been criticized for not accounting for the difference in spatial resolutions between GCMs and observations. The spatial distributions of $p_{fut} - p_{his}$ under RCP8.5 show that their differences between the site scale and the GCMs' original resolutions are marginal. Given the considerable
discrepancies among GCM outputs, the effects of spatial resolutions on projected changes are negligible.

## 1 Introduction





Under global warming, changes in extreme climate can lead to significant impacts on the occurrence and severity of natural disasters which will result in changes in risk of failure for urban infrastructures, such as reservoirs, highway, urban drainage systems, etc. (e.g. Easterling et al., 2000; IPCC, 2012; Trenberth et al., 2015; Li et al., 2016). To achieve long-term climate resilience, it is crucial to estimate of future climate extremes with an acceptable level of confidence as the scientific basis for

engineering design and policy making (e.g. O'Brien et al., 2004; Huntingford et al., 2013). Regional- and global- scale projections of precipitation extremes under various future climate scenarios have been extensively conducted based on Global Climate Models (GCMs) from Phases 3 and 5 of the Coupled Model Intercomparison Project (CMIP3) and (CMIP5) (e.g. IPCC, 2012; Li et al., 2013; Sillmann et al., 2013b). More frequent and severe precipitation extremes are consistently projected in most of the previous studies, which can be explained by the Clausius-Calpeyron relationship (C-C relationship;

Clausius 1850; Clapeyron 1834). According to the C-C relationship, a warmer atmosphere can hold more water vapor at an increase rate of 7%/K rise in temperature.

Although most previous studies agreed on projected increases in future precipitation extremes, they hardly made agreements on the changing rates because they were based on either different GCMs or resolutions. An assessment on performances of

GCMs in simulating precipitation extremes is important to understand the reliability of future projections, which can be achieved by comparing precipitation extremes derived from GCMs with reanalysis data (e.g. Sillmann et al., 2013a), remote sensing data (e.g. Dosio et al., 2015), or gridded or station-based observations (e.g. Nikulin et al., 2011; Li et al., 2013; Sillmann et al., 2013a). Sillmann et al. (2013a) compared precipitation extremes from CMIP3 and CMIP5 GCMs to those from four reanalysis datasets, including ERA40, ERA-Interim, NCEP/NCAR, and NCEP-DOE, as well as a gridded

observation-based dataset HadEX2. They concluded that CMIP5 models are able to simulate precipitation extremes in comparison with the gridded observations HadEX2 dataset, but the discrepancies between the reanalysis datasets and those between GCMs are considerable. The assessment over Europe conducted by Nikulin et al. (2011) found out that the biases in simulated precipitation extremes against observations have a complex and spotty spatial pattern, and biases tend to be substantial over mountains and areas with fewer observational stations.

It is difficult to directly compare GCMs' simulations against reference data from various sources partly because their resolutions vary widely (e.g. Wehner 2004; Hegerl et al., 2004). Further, comparing gridded GCM simulations against station-based observations can be problematic, particularly for precipitation extremes, because precipitation extremes were firstly proposed and defined at the site scale based on station-based observations, and it has been well recognized that

precipitation extremes are more spatially heterogeneous than temperature (Frich et al., 2002; Nikulin et al., 2011; Chen et al., 2016). Previous studies have acknowledged that intensities of precipitation extremes are sensitive to spatial resolutions (e.g. Kharin et al., 2005; Emori et al., 2005; Chen and Knutson, 2007). Kharin et al. (2005) estimated the averaged 24-h precipitation of 20-yr return period of a group of GCMs as a function of their horizontal grid resolutions, and found a significant reduction of 24-h precipitation of 20-yr return period as the grid cell size increases. From interpolating



0.25º×0.25º daily observed precipitation to resolutions ranging from 0.5º×0.5º to 4º×4º, Chen and Knutson (2007) also found considerable reductions of extreme precipitation indices at coarser resolutions compared to those at finer resolutions. Their results implied that GCMs at their raw resolutions, generally about 1º×1º to 3º×3º, tend to underestimate precipitation extremes at finer resolutions. These previous studies focused on the sensitivity of precipitation extreme values $p$ to spatial

resolutions $r$. However, how do spatial resolutions affect projections of changes in precipitation extremes, e.g. $p_{fut} - p_{his}$ (future extremes – historical extremes), is still unclear. Since $p_{fut} - p_{his}$ is a common metric to project future changes, therefore an investigation on its sensitivity to spatial resolutions is crucial to understand the uncertainties and reliability of GCMs' projections in precipitation extremes with reference to spatial resolutions.

In practice, downscaling (e.g. statistical and dynamical approaches) and upscaling (e.g. interpolation) are commonly adopted to either downscale coarse GCM simulations to a finer resolution or upscale reference data at a finer resolution to a coarser resolution (e.g. Mladjic et al., 2011; Eden et al., 2012). The advantages and shortcomings of various downscaling and upscaling methods have been extensively discussed and examined in previous studies (e.g. Schmidli et al., 2006; Xu and Yang, 2012). For example, statistical downscaling is usually criticized for lack of physical basis, while dynamical

downscaling, though computationally intensive, may result in more uncertainties to the results caused by the limitations of regional climate models (Wilby 1994; Wilby and Wigley 1997). In general, the strategies to solve the differences in spatial resolutions between GCMs and other data sources, especially site-scale observations, can be classified into three types: (1) downscale GCM outputs to the site scale, and compare downscaled simulations and station-based observations at the site scale (e.g. Li et al., 2013; Mullan et al., 2016); (2) upscale site-scale observations to grid cells of spatial resolutions of GCMs

and compare the results at a gridded scale (e.g. Sillmann et al., 2013a, 2013b); (3) compare grid-scale indices with site-scale observations directly under the assumption that site-scale observations are representative of areal distributed conditions (hereafter "grid-point comparison") (e.g. Ma et al., 2009; Hu et al., 2016). Because intensities of precipitation extremes are sensitive to spatial resolutions, projections of precipitation extremes at various spatial resolutions should be different. Here we examine the effect of spatial resolutions on changes in precipitation extremes between two periods, i.e. $p_{fut} - p_{his}$, based

on the site-scale observations and gridded GCM simulations across China. Therefore, this study aims to 1) quantify the sensitivity of $p_{fut} - p_{his}$ to spatial resolutions from site to gridded scales; 2) analyze performances of GCMs at various resolutions in simulating precipitation extremes using the three aforementioned comparison strategies, i.e. site-scale, grid-scale, and grid-point comparisons; and 3) examine impacts of spatial scales on projected changes in seasonal precipitation extremes across China given performances of GCMs at various scales. The results of this study will help us to better

understand the effect of spatial resolutions on projections of precipitation extremes, as well as the reliability and scaling-aggregation problems in GCM simulations.

**2 Data**





Ground observations of daily precipitation and temperature from 509 rain stations across China are collected from the National Meteorological Information Center of China Meteorological Administration (Fig. 1). The observed precipitation is valid for the period of 1960-2010 under good quality control (Zhang et al., 2011). Besides the lack of long-term ground observations in the Tibetan Plateau and the Taklimakan Desert in the westernmost China, the ground stations are fairly

evenly distributed across China. The gridded observations at the 2.5°×2.5° spatial resolution are constructed by interpolating the station-based observations using the natural neighbor interpolation method (Watson 1992, 1994; Conti et al., 2014). Furthermore, we collect simulated daily precipitation from 12 GCMs in the World Climate Research Program's CMIP5 for the historical, Representative Concentration Pathway 2.6 (RCP2.6) and RCP8.5 scenarios (Tab. 1; Van Vuuren et al. 2011; Riahi et al. 2011; Taylor et al. 2012). In the grid-scale comparison, GCM outputs are re-gridded to the 2.5°×2.5° spatial

resolution using the bilinear algorithm in Earth System Modeling Framework software so that GCM outputs have the same resolution of gridded observations (Hill et al., 2004). Precipitation extreme indices, including consecutive dry days (CDD), number of wet days (NW), max 5 day precipitation amount (R5d), number of heavy precipitation days (R10), precipitation fraction due to very wet days (R95T) and simple daily intensity index (SDII), are estimated for spring (MAM), summer (JJA), autumn (SON), and winter (DJF) (Tab. 2). These precipitation extremes have been extensively used in the literature

(e.g. IPCC, 2012; Sillmann and Roeckner, 2008; Li et al., 2013; Chen et al. 2016).

**3 Methodology**

GCM precipitation extremes at the site scale are computed by downscaling GCM outputs from their raw resolutions to the site scale based on a statistical downscaling method (Zhang et al., 2005; Li et al., 2011; Li et al., 2013). Firstly, the gridded

GCM simulations are smoothed by calculating the inverse distance weighted average of four neighboring grid cells. Secondly, transfer functions are developed to map gridded GCMs values to each target station. The inverse distance weight $w_i$ of a neighboring grid cell $i$ can be computed as:

$$w_i = \frac{(d_i + \varepsilon)^{-1}}{\sum_{i=1}^{4} (d_i + \varepsilon)^{-1}} \tag{1}$$

where $\varepsilon$ is a very small value to prevent division by zero, and $d_i$ represents the angular distance between the grid cell center

and the target station. The $d_i$ can be calculated by:

$$d_i = cos^{-1}[sin(c_{lat,i})sin(s_{lat}) + cos(c_{lat,i})cos(s_{lat})cos(c_{lon,i} - s_{lon})] \tag{2}$$

where $c_{lat,i}$ and $c_{lon,i}$ denote the latitude and longitude of the center of the grid cell $i$, respectively; $s_{lat}$ and $s_{lon}$ denote the latitude and longitude of the target station, respectively.

Then the smoothed GCM value of the target station $p_s$ can be computed as:

$$p_s = \sum_{i=1}^{n} w_i p_i \tag{3}$$





where $p_i$ denotes the value in the neighboring grid cell $i$, and $n$=1,2,3,4 is the number of available neighboring grid(s). Next, the smoothed GCM values $p_{s,train}$ and observed values at the target station $p'_{s,train}$ in the training period are paired based on their quantiles. We then identify the best fitting nonlinear and linear relationships between these pairs. These nonlinear and linear relationships, called transfer functions, can be used to transfer $p_{s,train}$ to $p'_{s,train}$. $p_{s,train}$ within the range of the

5 nonlinear function is transferred to $p'_{s,train}$ by the nonlinear function. $p_{s,train}$ out of the range of the nonlinear function is transferred to $p'_{s,train}$ by the linear function. These relationships are developed based on $p_{s,train}$ and $p'_{s,train}$ in the training period. The smoothed GCM values under the future scenarios $p_{s,fut}$ are then downscaled to the site-scale value $p'_{s,fut}$ based on these relationships.

Taylor diagrams are used to assess performances of GCMs in simulating precipitation extremes by comparison against the observation (Taylor, 2001; Kwok 2011; Li et al., 2013). In a Taylor diagram, the correlation ($R$), root-mean-square difference ($E^2$), and ratio of variances of the models ($\sigma_m$) and observation ($\sigma_o$) between simulations and observation are associated through the cosine relationship. The $E^2$ and $\sigma_m$ are normalized by the $\sigma_o$ as below:

$$\hat{E}^2 = (\frac{E}{\sigma_o})^2 \qquad \text{and} \qquad \hat{\sigma}_m = \frac{\sigma_m}{\sigma_o} \qquad\qquad\qquad (4)$$

Then the polar coordinates $\hat{\sigma}_m$ and $\cos^{-1}R$ of a GCM can locate a point in the Taylor diagram. Based on the cosine

relationship, the corresponding $\hat{E}$ is obtained and represented by the dashed semicircles centered at the unity on the abscissa in the diagram. After the normalization, the observation is plotted at the unit distance from the origin along the abscissa. The performance of a GCM can be described by the degree to proximity between the point representing the GCM and the point representing the observation. The 2-sample Kolmogorov-Smirnov (KS) test, a commonly used nonparametric

hypothesis test, is used to examine whether the probability distributions of downscaled and observed extremes are statistically identical (Massey, 1951; Miller, 1956; Stephens, 1970). The KS test is conducted at 5% significance level.

## 4 Results and discussions

### 4.1 Sensitivity of changes in precipitation extremes to spatial resolutions

Datasets of precipitation extremes at various spatial resolutions from the site scale to the 4°×4° gridded scale are constructed by aggregating daily precipitation of stations covered by grid cells centered at each of the 509 stations. For example, for a grid cell $i$ centered at a station $j$ with a cell size of 2.5°×2.5°, the daily precipitation of grid cell $i$ is the mean of precipitation in $n$ stations covered by that grid cell. We split the period of 1961-2005, the common years of observations and the GCM historical scenario, into a pseudo-future period 1986-2005 and a pseudo-historical period 1961-1985, so the change in

precipitation extremes between these two periods is $p_{1986-2005} - p_{1961-1985}$, which are estimated at gridded observations at





various spatial resolutions. Projections of future precipitation extreme were usually based on changes between a future period and a historical period in previous studies, i.e. $p_{fut} - p_{his}$. Therefore, analyzing the sensitivity of observed $p_{1986-2005} - p_{1961-1985}$ to various grid cell sizes will help us to better understand impacts of spatial resolutions on future projections of precipitation extremes.

The sensitivity analysis based on the observational data shows that spatial resolutions have considerable impacts on magnitudes of precipitation extremes in the pseudo-historical period, i.e. $p_{1961-1985}$, and the impacts to different precipitation extremes are various (Fig. 2). The magnitudes of CDD, R5d, R10, R95T, and SDII are significantly reduced as spatial resolutions are coarser, due to the averaging effect of precipitation which is highly variable spatially. R5d can

decrease from about 120 days to about 60 days (about a 50% reduction) when the spatial resolutions increase from the site scale to the 4º×4º resolution. SDII decreases by 40% at the 4º×4º resolution compared to the site-scale observations. CDD is around 50 days at the site scale, but decreases to 41 days at the 4º×4º resolution. Different from other extremes, NW is found to increase with coarser spatial resolutions which are expected. On average NW doubles at the 4º×4º resolution compared to those at the site scale. These changes in magnitudes of different precipitation extreme indices as the grid cell sizes increases

can be explained by the spatial heterogeneity of daily precipitation. Different regions in a grid cell can experience different storm mechanisms and precipitation characteristics, such as the occurrence, storm intensity, etc., therefore the probability of occurrence of precipitation events of a grid cell increases when the grid cell size increases. Hence NW increases but CDD decreases in coarser spatial resolutions. On the other hand, the spatially heterogeneity of precipitation intensity means that the intensity of heavy precipitation events in a location can be attenuated by the surrounding less intensive precipitation, thus

the average precipitation intensity decreases as a grid cell becomes larger, such as reductions in R5d, R10, R95T, and SDII as grid cell sizes increase. Our results for various precipitation extremes are in good agreement with the findings of Chan and Knutson (2007), in which substantial reductions in 30-yr return levels of daily precipitation was found at coarser spatial resolutions.

In contrast, the changes between precipitation extremes of 1986-2005 and 1961-1985, i.e. $p_{1986-2005} - p_{1961-1985}$, are much less sensitive to spatial resolutions (Fig. 2). For example, the site-scale SDII of 1986-2005 is found to be 0.2 mm higher than that of 1961-1985, while the difference decreases to about 0.05 mm at the 4º×4º spatial resolution. The difference of changes in SDII between the 4º×4º resolution and the site scale, i.e. $(SDII_{1986-2005,4º} - SDII_{1961-1985,4º}) - (SDII_{1986-2005,0º} - SDII_{1961-1985,0º})$, is about 0.15 mm, which is only 2.9% of the difference of magnitudes of SDII between the 4º×4º

resolution and the site scale, i.e. $SDII_{1961-1985,4º} - SDII_{1961-1985,0º}$, which is about 4 mm, showing that changes in SDII between two periods is less sensitive to spatial resolutions than its magnitudes to spatial resolutions. The insensitivity of changes between two periods with respect to spatial resolutions also can be identified for other precipitation extreme indices. The differences of changes in CDD, NW, R10, R5d, and R95T between the 4º×4º resolution and the site scale





$(p_{1986-2005,4^o} - p_{1961-1985,4^o}) - (p_{1986-2005,0^o} - p_{1961-1985,0^o})$ are about 13.3%, -0.7%, -28.9%, 0.0% and 0.0% of the differences in the magnitudes $p_{1961-1985,4^o} - p_{1961-1985,0^o}$, respectively.

The C-C relationship indicates that the water holding capacity of the atmosphere increases at about 7% per K rise in temperature, implying that precipitation extremes should change proportionally as temperature increases. To better understand the insensitivity of $p_{fut} - p_{his}$ to spatial resolutions, we define fractional changes in precipitation extremes relative to temperature changes across grid cells as slope $S$, which is calculated as

$$S = \left(\frac{p_{1986-2005} - p_{1961-1985}}{p_{1961-1985}}\right) / (T_{1986-2005} - T_{1961-1985}) \times 100\% \qquad (5)$$

where $T_{1986-2005}$ and $T_{1961-1985}$ are the mean temperature of pseudo-future period 1986-2005 and pseudo-historical period 1961-1985, respectively. Relationships of $S$ derived from observations and GCM simulations at various spatial resolutions are shown in Fig. 3a. The $S$ of NW, R5d, R95T, and SDII change slightly as spatial resolution increases at rates of about 0%/K, 2-3%/K, 2-3%/K, and 0-2%/K, respectively. At the same time, as expected, changes in temperature $T_{1986-2005} - T_{1961-1985}$ are much more spatially homogenous than precipitation as shown in Fig. S4. Therefore, based on Eq. 5, the stable S and $T_{1986-2005} - T_{1961-1985}$ at various resolutions mean $p_{1986-2005} - p_{1961-1985}$ should be relatively insensitive to spatial resolutions (Fig. 2). $S$ of R95T and R10 change more considerably with an increase from about 4%/K to 15%/K, and a decrease from about 0%/K to -5%/K as the spatial resolution increases from the site scale to the 4°×4° resolution, respectively. Therefore, sensitivities of changes in R95T and R10 are larger than the sensitivities of other extremes, but are still much smaller than the sensitivities of their magnitudes to spatial resolutions (Fig. 2). The dashed lines in Fig. 3 are regressions of slopes of precipitation extremes derived from GCMs versus their raw resolutions, showing that the slopes of GCMs change in similar rates with the observations.

Based on the above definition of slope $S$, the change in an extreme index between the future and historical periods at a spatial resolution $r$ can be computed by

$$p_{f,r} - p_{h,r} = S_r \times p_{h,r} \times (T_{f,r} - T_{h,r}) \qquad (6)$$

where $p_{f,r}$ and $p_{h,r}$ are precipitation extremes of future and historical periods at a spatial resolution $r$, respectively; $T_{f,r}$ and $T_{h,r}$ are the climatology of annual mean temperature of future and historical periods at resolution $r$, respectively. Therefore, the difference of changes in extremes of two periods between resolutions $r1$ and $r2$ can be computed as

$$(p_{f,r2} - p_{h,r2}) - (p_{f,r1} - p_{h,r1}) = S_{r2} \times p_{h,r2} \times (T_{f,r2} - T_{h,r2}) - S_{r1} \times p_{h,r1} \times (T_{f,r1} - T_{h,r1})$$

$$= \frac{\frac{S_{r2}}{S_{r1}} \times \frac{p_{h,r2}}{p_{h,r1}} \times \frac{T_{h,r2}}{T_{h,r1}} - 1}{\frac{p_{h,r2}}{p_{h,r1}} - 1} \times S_{r1} \times (T_{f,r1} - T_{h,r1}) \times (p_{h,r2} - p_{h,r1}) \qquad (7)$$

Based on Eq. 7, the difference of changes in extremes at resolutions $r1$ and $r2$, i.e. $(p_{f,r2} - p_{h,r2}) - (p_{f,r1} - p_{h,r1})$, can be related to the difference of magnitudes of extremes at resolutions $r1$ and $r2$, i.e. $p_{h,r2} - p_{h,r1}$. Because temperature is insensitive to spatial resolution (Fig. S1), then $\frac{T_{h,r2}}{T_{h,r1}} \approx 1$. To compare other spatial resolution with the site scale, we set r1 =





$0^o$. According to Fig. S1, for the pseudo-future period 1986-2005 and the pseudo-historical period 1961-1985, $T_{f,0^o} - T_{h,0^o} = 0.6248$ K. In this case, Eq. 7 can be simplified as

$$\left(p_{f,r2} - p_{h,r2}\right) - \left(p_{f,0^o} - p_{h,0^o}\right) = 0.6248 \times \left(\frac{S_{r2} \times \frac{p_{h,r2}}{p_{h,0^o}} - S_{0^o}}{\frac{p_{h,r2}}{p_{h,0^o}} - 1}\right) \times \left(p_{h,r2} - p_{h,0^o}\right) \qquad (8)$$

According to Figs. 2 and 3a, the range of $S_{r2}$ for the extreme indices investigated is about [-5, 15] in percentage, and the

range of $\frac{p_{h,r2}}{p_{h,0^o}}$ is about [0.5, 1.5]. Figs. 3b and 3c show the values of $0.6248 \times \left(\frac{S_{r2} \times \frac{p_{h,r2}}{p_{h,0^o}} - S_{0^o}}{\frac{p_{h,r2}}{p_{h,0^o}} - 1}\right)$ versus $S_{r2}$ and $\frac{p_{h,r2}}{p_{h,0^o}}$ when

$S_{0^o} = 0\%$ and $S_{0^o} = 5\%$. The equation approaches infinity when $\frac{p_{h,r2}}{p_{h,0^o}} = 1$. For most cases, $-0.2 < 0.6248 \times$

$\left(\frac{S_{r2} \times \frac{p_{h,r2}}{p_{h,0^o}} - S_{0^o}}{\frac{p_{h,r2}}{p_{h,0^o}} - 1}\right) < 0.3$, implying a much smaller sensitivity of $\left(p_{f,r2} - p_{h,r2}\right) - \left(p_{f,0^o} - p_{h,0^o}\right)$ to spatial resolution

compared to $p_{h,r2} - p_{h,0^o}$. Based on Eq. 8, the sensitivity of changes in precipitation extremes to spatial resolutions is a function of the sensitivity of the magnitudes to spatial resolutions, the sensitivity of the slopes to spatial resolutions, and

changes in temperature. According to this equation, Tab. 3 compares the estimated and observed sensitivity of changes and magnitudes of precipitation extremes between the site scale and the 4°×4° resolution. The good agreement between the estimated and observed sensitivity demonstrates the skills of Eq. 8 to quantify the sensitivity of changes in precipitation extremes to spatial resolution, and explains the negligible changes in precipitation extremes between two periods at various spatial resolutions.

## 4.2 Performances of GCMs in simulating precipitation extremes at various resolutions based on the site-scale, grid-scale, and grid-point comparisons

According to the above analysis, magnitudes of precipitation extremes are very sensitive to spatial resolutions, while changes in precipitation extremes are much less sensitive, which may affect the assessment of precipitation extremes at various

spatial scales. An analysis about the effects of spatial resolutions on the assessment of precipitation extremes will be useful to compare assessments of precipitation extremes in previous studies conducted at different spatial resolutions. In this part, we assess seasonal precipitation simulated by GCMs against observations of 1991-2005 based on the site-scale, the grid-scale, and grid-point comparisons. In the site-scale comparison, GCM precipitation extremes are statistically downscaled from their raw resolutions to the 509 ground stations. In the grid-scale comparison, the station-based observations are

interpolated to grids of the 2.5°×2.5° resolution, and all GCMs are also re-gridded to the 2.5°×2.5° resolution. In the grid-point comparison, precipitation extremes estimated from GCMs at their original resolutions are directly compared to the site observations. Different downscaling and re-gridding algorithms should have significant impacts on the results obtained from the comparisons, but in this study, we focus on the performances of GCMs in simulating the magnitudes of precipitation extremes at different resolutions and how their performances based on magnitudes affect projected changes in precipitation





extremes at various spatial resolutions (see Section 4.3). Therefore, the effects of choices of downscaling and re-grid algorithms are not discussed in this study.

In the site-scale comparison, the statistical relationships between the grid- and site-scale precipitation extremes are developed during the period of 1961-1990. The relationships are then applied to downscale the grid-scale extremes of 1991-2005 to site-scale values. The multimodel ensemble mean of the downscaled precipitation extremes are then compared with ground observations. Taylor diagrams of spring, summer, autumn, and winter CDD show downscaled CDD match the ground observations well with a correlation of about 0.99 and a normalized standard deviation of about 1 (Fig. 4). Downscaled NW, R5d, R10, and SDII also agree well with observed data as shown in Taylor diagrams (Figs. S2-S4, Fig. 6), except the downscaled R95T is not as good, with correlation coefficients of about 0.4-0.5, and the normalized standard deviations about 0.6-0.8 (Fig. 5). The difference between the areal mean of downscaled precipitation extremes and observations across the rain stations at the site scale, i.e. $p_{GCM} - p_{obs}$, is shown in Tab. 4. Differences between downscaled and observed CDD for the four seasons are between −1.38 and 0.29 days. Downscaled CDD in spring and summer is slightly larger than the observations, with overestimation of 0.09 and 0.29 days, respectively. The statistical downscaling underestimates the autumn and winter CDD by 1.38 and 0.94 days, respectively. The downscaled R10 agrees well with the observations with differences of 0.09, -0.42, 0.35, and -0.23 days in spring, summer, autumn, and winter, respectively. The downscaled R5d are 0.37 mm higher, 4.55 mm lower, 2.32 mm higher, and 1.31 higher than the observations in spring, summer, autumn, and winter, respectively. The statistical downscaling method tends to underestimate R95T. The downscaled R95T is 0.26%, 1.74%, and 2.56% lower than the observations in spring, summer and winter, respectively. The performance of downscaled SDII is good, as indicated by the differences of 0.01 mm/day, -0.51 mm/day, 0.00 mm/day, and -0.19 mm/day in spring, summer, autumn and winter, respectively. Moreover, the 2-sample KS test is used to compare the probability distributions of downscaled with those of observed precipitation extremes. The proportions of stations with statistically the same probability distributions between downscaled and observed extremes are shown in Tab. 5. According to the 2-sample KS test at the 5% significance level, more than 96% of the stations have probability distributions of downscaled and observed CDD, NW, R5d, and SDII that are statistically the same. In spring, summer, and autumn, more than 86% of the stations are identified with the same distributions of downscaled and observed R10, while the corresponding proportion for winter is relatively lower at 71%. The downscaled R95T performs better in summer than other seasons, when the proportions or stations that have statistically the same distributions are 79%, 76%, and 48% in spring, autumn, and winter, respectively. Fig. 7 shows the box-and-whisker plots of precipitation extremes derived from GCMs and observations based on the site-scale, grid-scale, and grid-point comparisons. Similar to the above discussion, in the site-scale comparison, areal means of simulated seasonal precipitation extremes are close to those of observations during 1991-2005, although simulations of summer R5d, R10, R95T, and SDII slightly underestimate the observations (Fig. 7). Therefore, the site-scale comparison indicates that the downscaled GCMs are able to simulate seasonal precipitation extremes with acceptable performance,



especially CDD, NW, R5d, R10 and SDII. Different downscaling methods can lead to various impacts on the site-scale comparison, but this is not the major scope of the study.

In the grid-scale comparison, all GCM daily precipitation outputs are firstly re−gridded to the 2.5°×2.5° spatial resolution

from which precipitation extreme indices are estimated. The performances of GCMs in simulating precipitation extremes in the grid-scale comparison are not as good as those in the site-scale comparison. At the 2.5°×2.5° spatial resolution, the correlation of ensemble CDD is around 0.8, while the normalized standard deviations vary from 0.6 to 1.4 (Fig. 4). The correlations of R95T is about 0.8 in spring and summer, and about 0.5-0.7 in autumn and winter, and the normalized standard deviations are about 0.5-0.7 in spring and winter (Fig. 5). The correlations between simulated and observed grid-scale SDII

are mostly about 0.6-0.7, which are substantially lower than those at the site scale of about 0.95-0.99 (Fig. 6). In comparing areal means between simulated and observed extremes at the gridded scale, CDD, R95T, and SDII are all underestimated in the four seasons (Fig. 7). The simulated NW, R5d, and R10 are generally higher than the observations. Tab. 4 shows that differences between areal means of the GCMs and observations at the gridded scale are larger than those at the site scale. The site-scale and grid-scale comparisons show that GCMs generally have the ability in simulating representative seasonal

precipitation extremes, and their performances at the site scale are much better than at the gridded scale. The better performances of GCMs at the site scale are attributed to the statistical downscaling method which explicitly incorporates the site-scale information of precipitation to the future values when downscaling the GCMs, while re-gridding GCM simulations based on certain interpolation algorithms does not incorporate this information.

In the grid-point comparison, precipitation extremes estimated from GCMs at their raw gridded resolutions are directly compared with the observed values at stations located within the corresponding grid cells. The Taylor diagrams show that the grid-scale CDD, R95T and SDII estimated at GCMs' original resolution hardly match observed values at the site scale (Figs. 4-6). For R95T, the correlations between GCMs and observations in four seasons are negative, and the normalized standard deviation of the multimodel ensemble is 2. The normalized standard deviations of CDD in spring, autumn, and winter are

about 0.5-0.6, and those in summer is better with value of 1. The normalized standard deviations of SDII in four seasons are only about 0.4-0.6, which is much lower than the downscaled SDII. The correlations of CDD and SDII between at GCMs raw resolutions and at the site scale are mostly around 0.6-0.7, which are lower than the downscaled CDD and SDII with correlations of 0.95-0.99. Tab. 4 shows the $p_{GCM} - p_{obs}$ in the grid-point comparison are larger than those in the site-scale comparison but similar to those in the grid-scale comparison. The CDD derived from GCMs' original resolutions are 7.55,

3.68, 5.80, and 10.09 days less than the site-scale observations in spring, summer, autumn, and winter, respectively. On the other hand, GCMs at raw resolutions overestimate NW and R10, especially for gridded NW, which are 14.52, 22.82, and 11.52 days larger than the site-scale observations in spring, summer, and autumn, respectively. The gridded R5d is 7.31 mm higher, 16.29 mm lower, and 6.01 mm higher than the grounded observation in spring, summer, and winter, respectively. The box-and-whisker plots of the second and third columns in Fig. 7 show that similar results are obtained from comparing





simulated with observed precipitation extremes in the grid-scale and grid-point comparisons. Therefore, the grid-point comparison which directly compares gridded extremes with the site-scale values may be comparable to the grid-scale comparison which re-grid all GCMs and observations to the 2.5°×2.5° spatial resolution, although the resolutions of GCMs and observations are varying in the grid-point comparison.

The magnitudes of precipitation extremes in the site-scale, grid-scale and grid-point comparisons in the box-and-whisker plots in Fig. 7 show similar changes as in Fig. 2 for data of coarser resolutions. In the grid-scale comparison, the magnitudes of CDD, R5d, and SDII estimated from both GCMs and observations at the 2.5°×2.5° spatial resolution are smaller than those at the site scale, but NW are larger than those at the site scale. As expected, comparison results for precipitation
extremes are dependent on spatial resolutions.

We further analyze the spatial distributions of the differences between simulated and observed extremes, i.e. $p_{GCM} - p_{obs}$, obtained from the site-scale and grid-point comparisons. In the grid-point comparison, for each individual GCM, the GCM precipitation extreme of each station is the value of the grid cell in which the station is located. The precipitation extremes
based on values given in GCMs' grid cells where the stations are located are compared against the observed extremes of the corresponding stations. Based on the above discussion, results for the grid-scale comparison are similar with that of the grid-point comparison, and so only the latter is considered in the following spatial analysis. As shown in Fig. 8, at the site scale, the simulated CDD in most parts of China is within +/- 2 days compared to the observed CDD in spring and summer. The observed CDD in autumn are 2-6 days higher than that of the GCM simulations, especially for the southeast and northwest
China, while the winter CDD in northwest China are overestimated by GCMs by about 4-8 days. In the grid-point comparison, GCMs at the original spatial resolutions significantly underestimate the CDD, e.g. the CDD simulated by GCMs in middle and western China is 8 days less than that of the observations. At the site scale, the differences between simulated and observed R5d are within -5 to 5 mm in most parts of China, but the summer R5d in the southeast China is underestimated by 10 – 20 mm in the GCMs. In autumn, GCMs overestimates the R5d in southeast China by 5 – 15 mm. In
the grid-point comparison, the bias of GCM R5d relative to observed R5d has an obvious spatial pattern. The spring and summer R5d in the southeast China simulated by the GCMs are 20 mm less than the observations, while those in southwest China and the Yangtze River Basin are >20 mm higher than the observations. The differences between the simulated and observed SDII are within -1 to 1 mm/day across China at the site scale, which demonstrates the good performance of downscaled GCM simulations in representing SDII. Similar to other indices, the differences between simulated and observed
SDII in the grid-point comparison are larger than those at the site scale, as shown by the underestimation of spring, summer and autumn SDII in southeast China in GCMs. In winter, the SDII simulated by GCMs in north China is about 1-3 mm higher than the observations. For NW and R95T, relatively large bias is found between GCMs and observations in west China compared to other regions, and in the grid-point assessment compared to the site-scale assessment (Fig. S5).



In the above discussion, we assess the magnitudes of precipitation extremes derived from GCMs and observations based on the site-scale comparison by downscaling the GCM extremes, the gird-scale comparison by interpolating GCMs and observations to the 2.5º×2.5º resolution, and the grid-point comparison by directly comparing GCMs at the original resolutions to the site-scale observations. The magnitudes of precipitation extremes in various comparison strategies are

different, and the directions of changes follow the sensitivity of precipitation extremes to spatial resolution as shown in Fig. 2. Performances of GCMs in simulating precipitation extremes in the site-scale comparison outperform those in the grid-scale and grid-point comparisons. This is expected because statistical downscaling in the site-scale comparison develops the empirical relationships between grid- and site- extremes which help to improve the performances of GCMs by reducing biases of GCMs. The performances of GCMs in the grid-scale comparison and grid-point comparison are somewhat

comparable, even though GCMs have better agreement with the observations in the grid-scale comparison.

## 4.3 Projected changes of seasonal precipitation extremes across China at the site scale and the GCMs' original resolutions

Given the understanding of differences found in magnitudes of precipitation extremes in the site-scale, grid-scale and grid-

point comparisons, in this section, we further examine whether significant differences can be found in the projected changes in precipitation extremes between different spatial resolutions, i.e. $(p_{f,r2} - p_{h,r2}) - (p_{f,r1} - p_{h,r1})$. Projected changes of precipitation extremes during 2071-2100 under RCP8.5 relative to 1971-2000 under historical scenario are conducted at the site scale and at the GCMs' raw spatial resolutions. Based on Eq. 7, differences in projected changes at various spatial resolutions will be positively related to projected temperature changes. Since temperature is projected to increase

substantially at the end of the 21st century under RCP8.5, this scenario is chosen so that the differences in projected changes with respect to resolutions will be more obvious. To display projected changes in precipitation extremes estimated at various raw spatial resolutions of GCMs which can be different from each other (Tab. 1), the precipitation extreme of each grid cell is assigned to the meteorological stations that are located with the grid cell. Then for each station, the multimodel ensemble of the precipitation extremes from different GCMs is computed as the ensemble mean of precipitation extremes from the

GCMs.

As predicted by the Eq. 7, differences in projected changes of seasonal precipitation extremes at the site scale and at their GCMs' raw resolutions are expected to be small (Fig. 9). At the site scale, CDD in the southeast China is projected to increase marginally by 2 – 4 days in spring and summer, and by 2 – 6 days in autumn, respectively. However, CDD is

projected to decrease in northwest China by the end of the 21st century, especially in spring and winter when the projected decrease can exceed 10 days. Projections of CDD from GCMs' original resolutions show very similar spatial patterns and change rates, while changes in winter are relatively modest compared to the site scale. At the site scale, decreases in NW in southeast China can be found in spring, summer, and winter with rates at about -4 - -1 days, while increase can be found in west China in spring and winter with rates at about 1-3 days. Again, the projected changes in NW at the GCMs' raw



resolutions and at the site scale show very similar patterns. At both scales, R5d is projected to increase across China, especially in southeast China. In spring, the R5d is projected to increase by 20 – 35 mm in southeast China, and by 0 – 15 mm in north. In summer, substantial increases in R5d are projected in east China, which are mostly larger than 40 mm. On the other hand, in northwest China, the summer R5d is only 0 - 5 mm higher than that during 1971-2000. In winter, R5d is merely projected to increase by 0 - 5 mm across China, which means only marginal change in winter R5d is expected. However, projected changes in the summer R5d at the raw resolutions of GCMs exhibit a different spatial pattern from those at the site scale. The summer R5d at GCM's raw resolutions reach the maximum in southwest China with increasing rate > 40 mm. In southeast China, R5d at the GCMs' raw resolutions is projected increase by 15 – 25 mm, smaller than those at the site scale. At both the site scale and GCMs' original resolutions, SDII is projected to increase in most parts of China in spring, summer and autumn by 1 – 2.5 mm.

Therefore, even though the differences in magnitudes of precipitation extremes are large between the site scale and the GCMs' raw resolutions, the differences in projected changes are much smaller between different scales, which is in agreement with the Eq. 7 and the sensitivity analysis. Statistical downscaling can significantly improve the performances of GCMs in projecting magnitudes of precipitation extremes (Figs. 4-7), but it does not seem to have considerable effect on the projected changes in precipitation extremes. Hence, given considerable degree of uncertainties among GCMs as shown in the Taylor diagrams (Figs. 4-6), the relatively minor effects of spatial resolutions on projected changes indicate that projections of changes in precipitation extremes at various spatial resolutions published in the previous studies should be comparable to some extent.

## 5 Conclusions

In this study, magnitudes $p$ and changes, i.e. $p_{fut} - p_{his}$, in precipitation extremes are estimated from the site scale to the grid scale with the aims to analyze the sensitivity of changes in precipitation extremes to spatial resolutions. Precipitation extremes derived from GCMs are evaluated against observations at the three commonly used strategies, i.e. the site-scale comparison after downscaling, the grid-scale comparison at the 2.5°×2.5° resolution, and the grid-point comparison that directly assesses precipitation extremes at GCMs' raw resolutions relative to the site-scale observations. Afterward, projected changes in seasonal precipitation extremes across China at different spatial resolutions are analyzed to investigate the effects of spatial resolutions to the projected changes. Based on the results and discussions of this study, the following conclusions can be drawn:

1) Magnitudes of precipitation extremes are sensitive to spatial resolutions, but changes in precipitation extremes between two periods are much less sensitive to spatial resolutions. As grid cell sizes increase, CDD, R5d, R10, R95T, and SDII tend to substantially decrease but NW tend to increase. However, changes in precipitation extremes are projected to be much more modest. For example, SDII of 1961-1985 decreases by 40% at the 4°×4° resolution compared to that at the site scale,



while the change in SDII between 1986-2005 and 1961-1985 are only about 3% of the decrease in the magnitude of SDII estimated at the site scale to the $4^\circ \times 4^\circ$ resolution.

2) The difference in changes in precipitation extremes of two periods at two resolutions is a function of the sensitivity of magnitudes of precipitation extremes to spatial resolutions, the sensitivity of fractional changes in precipitation extremes relative to temperature increase to resolutions, and changes in temperature of the two periods. The fractional changes relative temperature of NW, R5d, R95T and SDII are about 0-2%/K at the site scale. Our results demonstrate the insensitivity of changes in precipitation extremes to spatial resolutions, even though the magnitudes of precipitation extremes are sensitive to spatial resolutions.

3) Performances of GCMs in simulating precipitation extremes in the site-scale comparison are better than those in the grid-scale and grid-point comparisons, because of the empirical relationships developed by statistical downscaling between grid- and site-scale precipitation extremes. In the site-scale comparison, when the future extremes of GCMs are downscaled to the site scale, the relationship implicitly corrects model biases and other differences between models and observations. Furthermore, the grid-point comparison provides comparable assessment results with the grid-scale comparison, even though the grid-point comparison does not account for the problems of resolution difference between gridded GCMs and site observations, because interpolating site-scale observations to the grid scale hardly increases the representation of the areal climatic conditions constrained by the limited observations.

4) Although the magnitude of precipitation extremes at various resolutions are different, and the performances of GCMs at the site scale have been significantly improved after statistical downscaling compared to those at their original resolutions, projected changes in precipitation extremes between 2071-2100 under RCP8.5 and the historical period at the site scale and at GCMs' original resolutions are similar. This can be explained by the insensitivity of changes in precipitation extremes between two periods to spatial resolutions. Given the considerable degree of uncertainties in climate projections of different GCMs, the effects of spatial resolutions on projected changes play a relatively minor role. In other words, GCMs' projections of changes in precipitation extremes at various spatial resolutions in the previous studies are comparable to each other to some extent.

5) Under RCP8.5, R5d and SDII in the four seasons are projected to increase at the end of the 21$^{st}$ century across China. The projected increases are especially substantial in southeast China in summer with an increase larger than 40 mm for R5d and larger than 3.5 mm/day for SDII, respectively. Considerable increases in CDD are projected over the southeast China in autumn and winter, which exceeds 4 days at both the site scale and GCMs' original resolutions.

**Acknowledgements.** This work was substantially supported by research grants from the Research Grants Council of the Hong Kong Special Administrative Region, China (No. HKBU22301916 and No. CUHK441313), the Faculty Research Grant from Hong Kong Baptist University (No. FRG2/15-16/043), and National Science Foundation for Distinguished Young Scholars of China (No. 51425903). Observed daily precipitation and temperature are available at the National Meteorological Information Center for the China Meteorological Administration at




http://www.cma.gov.cn/2011qxfw/2011qsjgx/. Detailed information of the data can be obtained by contacting to the corresponding author at jianfengli@hkbu.edu.hk.

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



**Table 1. GCMs from CMIP5**

| GCM | Modeling Center | Resolution (Lon×Lat) | Data duration Historical | RCP8.5 |
|---|---|---|---|---|
| BCC-CSM1.1(m) | BCC | 320×160 | 1850-2012 | 2006-2100 |
| BNU-ESM | GCESS | 128×64 | 1950-2005 | 2006-2100 |
| CanESM2 | CCCma | 128×64 | 1850-2005 | 2006-2100 |
| CCSM4 | NCAR | 288×192 | 1850-2005 | 2006-2300 |
| CESM1(CAM5) | NSF-DOE-NCAR(1) | 288×192 | 1850-2005 | 2006-2100 |
| CNRM-CM5 | CNRM-CERFACS | 256×128 | 1850-2005 | 2006-2100 |
| CSIRO-Mk3.6.0 | CSIRO-QCCCE | 192×96 | 1850-2005 | 2006-2300 |
| GFDL-CM3 | NOAA GFDL | 144×90 | 1860-2005 | 2006-2100 |
| IPSL-CM5A-MR | IPSL | 144×143 | 1850-2005 | 2006-2100 |
| MIROC5 | TUT NIES JAMEST | 256×128 | 1850-2012 | 2006-2100 |
| MPI-ESM-MR | MPI-M | 192×96 | 1850-2005 | 2006-2100 |
| MRI-CGCM3 | MRI | 320×160 | 1850-2005 | 2006-2100 |



**Table 2. Definitions of precipitation extreme indices**

| Indices | Name | Definition | Unit |
|---|---|---|---|
| CDD | Consecutive dry days | Maximum number of consecutive days with daily precipitation < 1 mm | day |
| NW | Number of wet days | Number of days with daily precipitation $\geq$ 1 mm | day |
| R5d | Max 5 day precipitation amount | Maximum total 5 day precipitation | mm |
| R10 | Number of heavy precipitation days | Number of days with daily precipitation $\geq$ 10 mm | day |
| R95T | Precipitation fraction due to very wet days | Fraction of total precipitation due to events exceeding the 95[th] percentile of wet day (daily precipitation > 1 mm) during 1961-1990 | % |
| SDII | Simple daily intensity index | Annual total precipitation divided by the number of wet days (daily precipitation > 1 mm) in the year | mm/day |





**Table 3. Proportion of differences in changes in precipitation extremes to differences in magnitudes of precipitation extremes between the site scale and the 4°×4° resolution, i.e. $\frac{(p_{f,4^o}-p_{h,4^o})-(p_{f,0^o}-p_{h,0^o})}{p_{h,4^o}-p_{h,0^o}}$, estimated from Eq. 8 and from observations**

|  | $S_{0^o}$ (%) | $S_{4^o}$ (%) | $\frac{p_{4^o}}{p_{0^o}}$ | Estimated proportion (%) | Observed proportion (%) |
|---|---|---|---|---|---|
| CDD | -0.21 | -4.01 | 0.85 | 13.68 | 13.32 |
| NW | 0.18 | 1.10 | 1.87 | 1.36 | -0.65 |
| R5d | 2.22 | 2.46 | 0.53 | 1.20 | -0.01 |
| R10 | 4.62 | 12.09 | 0.92 | -49.98 | -28.86 |
| R95T | 1.53 | 4.53 | 0.71 | -3.63 | 0.01 |
| SDII | 2.31 | 0.89 | 0.53 | 2.45 | 2.88 |





**Table 4. Difference of areal means of seasonal precipitation extremes derived from GCMs and those ground observations across China, i.e. $p_{GCM} - p_{obs}$, at the site-scale, grid-scale, and grid-point comparisons**

|  | Indices | Spring | Summer | Autumn | Winter |
|---|---|---|---|---|---|
| Site-scale | CDD (day) | 0.09 | 0.29 | -1.38 | -0.94 |
|  | NW (day) | 0.28 | -0.07 | 1.18 | -0.55 |
|  | R5d (mm) | 0.37 | -4.55 | 2.32 | 1.31 |
|  | R10 (day) | 0.09 | -0.42 | 0.35 | -0.23 |
|  | R95T (%) | -0.26 | -1.74 | 0.34 | -2.56 |
|  | SDII (mm/day) | 0.01 | -0.51 | 0.00 | -0.19 |
| Grid-scale | CDD (day) | -5.71 | -1.28 | -3.78 | -9.86 |
|  | NW (day) | 13.35 | 14.47 | 9.61 | 5.08 |
|  | R5d (mm) | 11.08 | -4.57 | 4.79 | 5.75 |
|  | R10 (day) | 1.42 | 1.12 | 1.09 | 0.07 |
|  | R95T (%) | -0.13 | -0.10 | -0.08 | -0.23 |
|  | SDII (mm/day) | -0.35 | -2.06 | -0.53 | 0.34 |
| Grid-point | CDD (day) | -7.55 | -3.68 | -5.80 | -10.09 |
|  | NW (day) | 14.52 | 22.82 | 11.52 | 4.84 |
|  | R5d (mm) | 7.31 | -16.29 | 0.89 | 6.01 |
|  | R10 (day) | 1.71 | 1.59 | 1.20 | 0.27 |
|  | R95T (%) | 0.28 | 0.12 | 0.34 | 0.51 |
|  | SDII (mm/day) | -1.51 | -5.36 | -2.02 | 0.14 |



**Table 5. Proportion (%) of stations with the same probability distributions of downscaled and observed precipitation extremes among the 509 ground stations based on the 2-sample KS test**

| Indices | Spring | Summer | Autumn | Winter |
|---|---|---|---|---|
| CDD (day) | 99 | 97 | 98 | 97 |
| NW (day) | 99 | 98 | 97 | 96 |
| R5d (mm) | 99 | 99 | 98 | 99 |
| R10 (day) | 86 | 91 | 86 | 71 |
| R95T (%) | 79 | 91 | 76 | 48 |
| SDII (mm/day) | 99 | 98 | 98 | 96 |



**Figure Caption List:**

Figure 1. Locations of meteorological stations across China.

Figure 2. Average magnitudes of 1961-2005 $p_{1961-1985}$ (orange) and changes between 1986-2005 and 1961-1985 $p_{1986-2005} - p_{1961-1985}$ (blue) of CDD (day), NW (day), R5d (mm), R10 (day), R95T (%), and SDII (mm/day) at various spatial resolutions in degree.

Figure 3. (a) Slope of fractional changes of precipitation extremes versus differences in the climatology of near-surface air temperature (%/K) between 1986-2005 and 1961-1985 at various spatial resolution. The solid lines are slopes of observations at various resolutions. The dashed lines are regressions of slopes of extremes estimated from the GCMs at their raw resolutions. (b) Values of $0.6248 \times \left( \dfrac{S_{r2} \times \frac{p_{h,r2}}{p_{h,0^o}} - S_{0^o}}{\frac{p_{h,r2}}{p_{h,0^o}} - 1} \right)$ versus $S_{r2}$ and $\dfrac{p_{h,r2}}{p_{h,0^o}}$ when $S_{0^o} = 0\%$. (c) the same as (b) but for $S_{0^o} = 5\%$.

Figure 4. Taylor diagrams of seasonal CDD derived from GCMs against observations based on the site-scale, grid-scale, and grid-point comparisons.

Figure 5. Taylor diagrams of seasonal R95T derived from GCMs against observations based on the site-scale, grid-scale, and grid-point comparisons.

Figure 6. Taylor diagrams of seasonal SDII derived from GCMs against observations based on the site-scale, grid-scale, and grid-point comparisons.

Figure 7. Box-and-whisker plots of areal mean of CDD (day), NW (day), R5d (mm), and SDII (mm/day) across China of observations, historical, RCP2.6 and RCP8.5 scenarios at the site-scale (the first column), grid-scale (the second column), and grid-point comparisons (the third column). Precipitation extremes of observations and historical scenario are during the period of 1991-2005, and those of RCP2.6 and RCP8.5 scenarios are during the period of 2071-2100.

Figure 8. Spatial distribution of difference of simulated precipitation extremes relative to observations, i.e. $p_{GCM} - p_{obs}$, for CDD (day), R5d (mm), and SDII (mm/day) during 1991-2005 at the site-scale and grid-point comparison.

Figure 9. Projected changes in (a) CDD (day), (b) NW (day), (c) R5d (mm), and (d) SDII (mm/day) during 2071-2100 under RCP8.5 relative to during 1971-2000 under the historical scenario at the site scale and at GCMs' original spatial resolutions.





**Figure 1. Locations of meteorological stations across China.**





Figure 2. **Average magnitudes of 1961-2005 $p_{1961-1985}$ (orange) and changes between 1986-2005 and 1961-1985 $p_{1986-2005} - p_{1961-1985}$ (blue) of CDD (day), NW (day), R5d (mm), R10 (day), R95T (%), and SDII (mm/day) at various spatial resolutions in degree.**





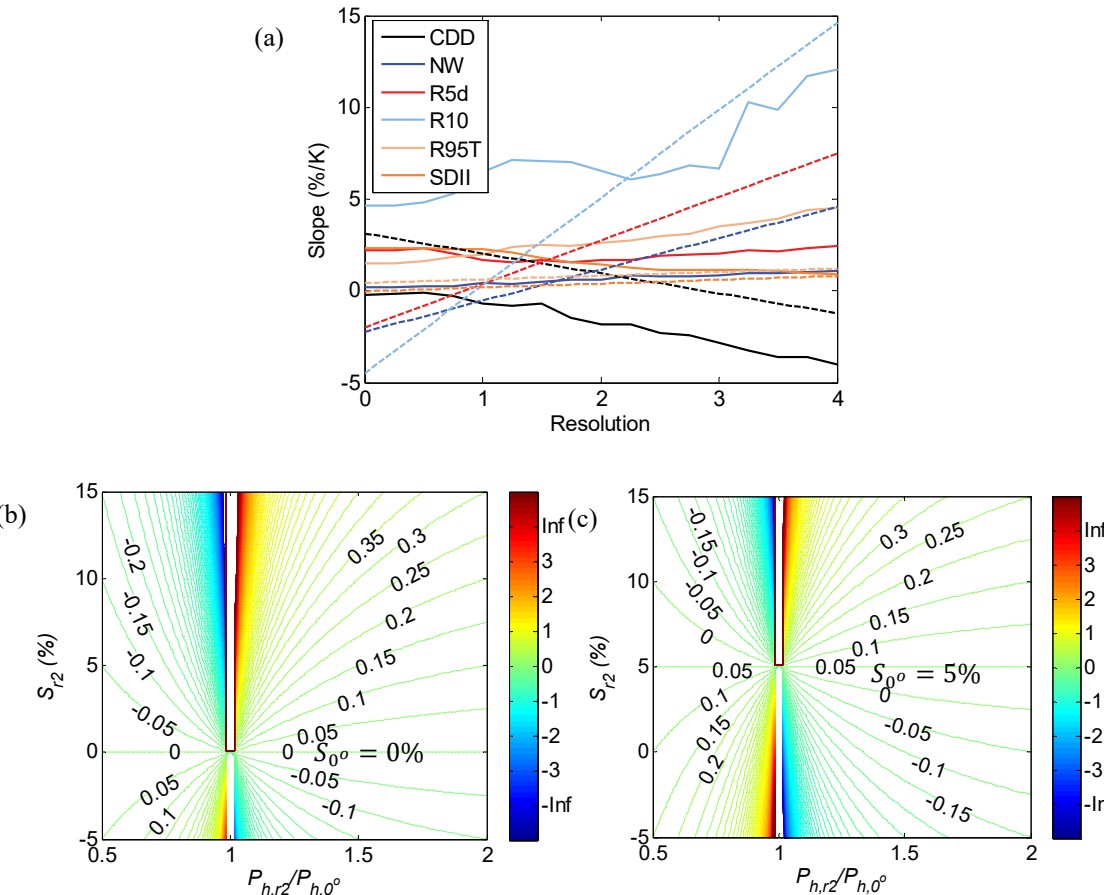

**Figure 3. (a) Slope of fractional changes of precipitation extremes versus differences in the climatology of near-surface air temperature (%/K) between 1986-2005 and 1961-1985 at various spatial resolution. The solid lines are slopes of observations at various resolutions. The dashed lines are regressions of slopes of extremes estimated from the GCMs at their raw resolutions. (b) Values of $0.6248 \times \left( \dfrac{S_{r2} \times \frac{p_{h,r2}}{p_{h,0^o}} - S_{0^o}}{\frac{p_{h,r2}}{p_{h,0^o}} - 1} \right)$ versus $S_{r2}$ and $\dfrac{p_{h,r2}}{p_{h,0^o}}$ when $S_{0^o} = 0\%$. (c) the same as (b) but for $S_{0^o} = 5\%$.**





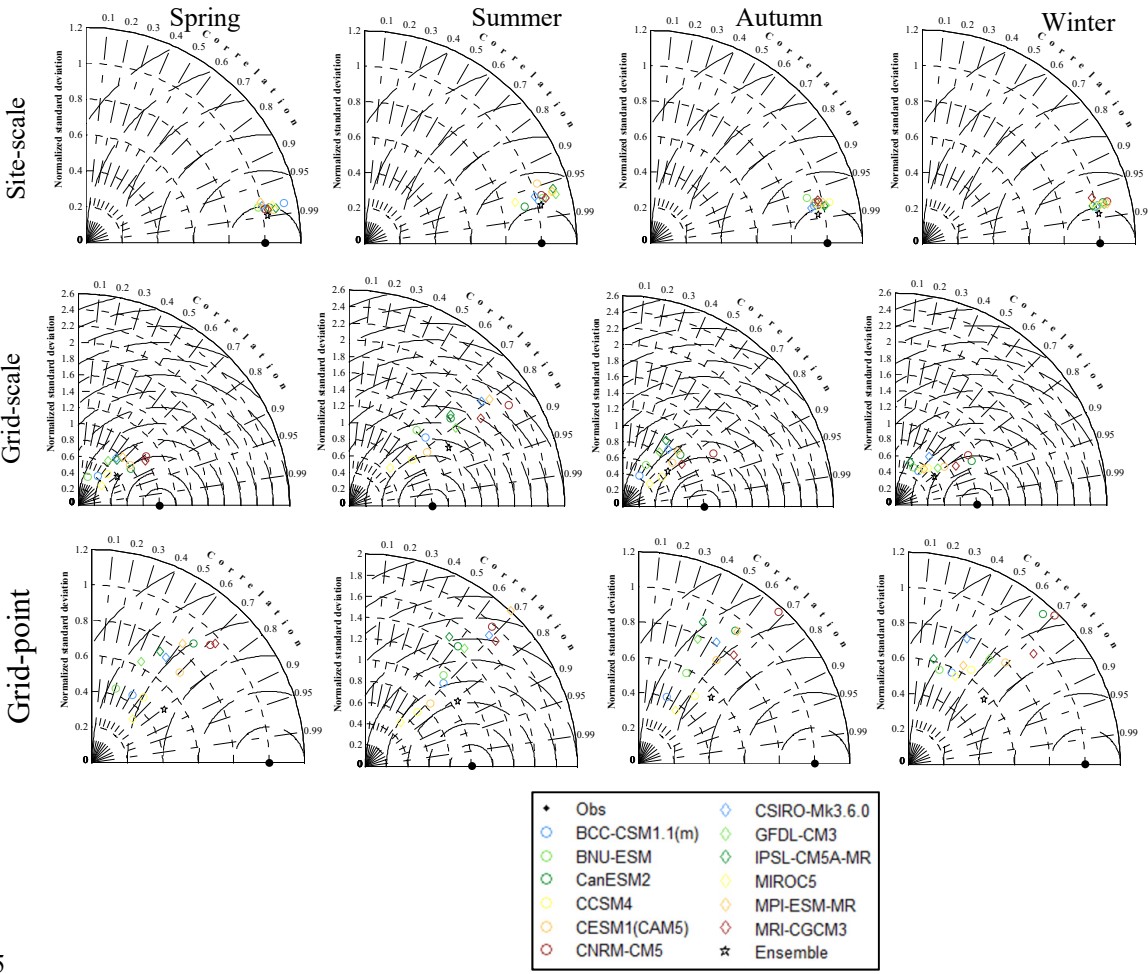

**Figure 4. Taylor diagrams of seasonal CDD derived from GCMs against observations based on the site-scale, grid-scale, and grid-point comparisons.**




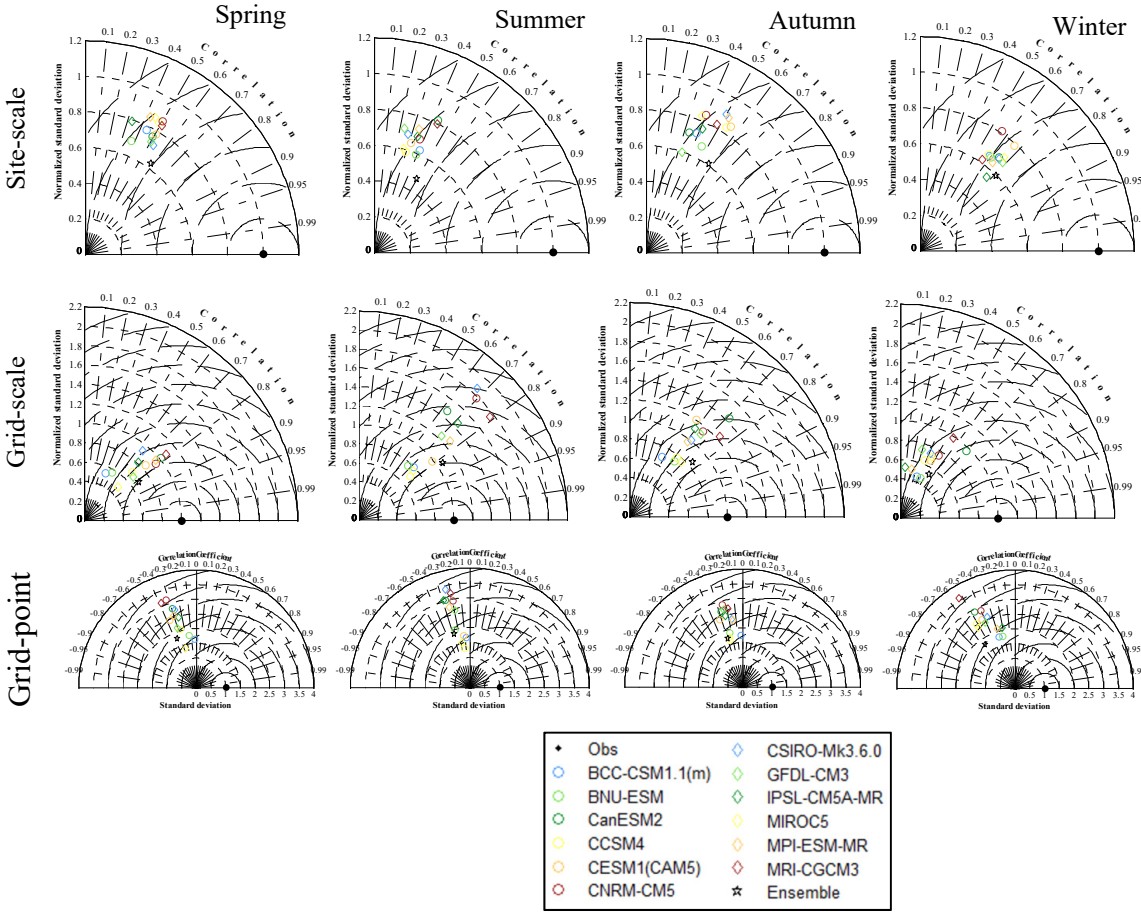

**5    Figure 5. Taylor diagrams of seasonal R95T derived from GCMs against observations based on the site-scale, grid-scale, and grid-point comparisons.**





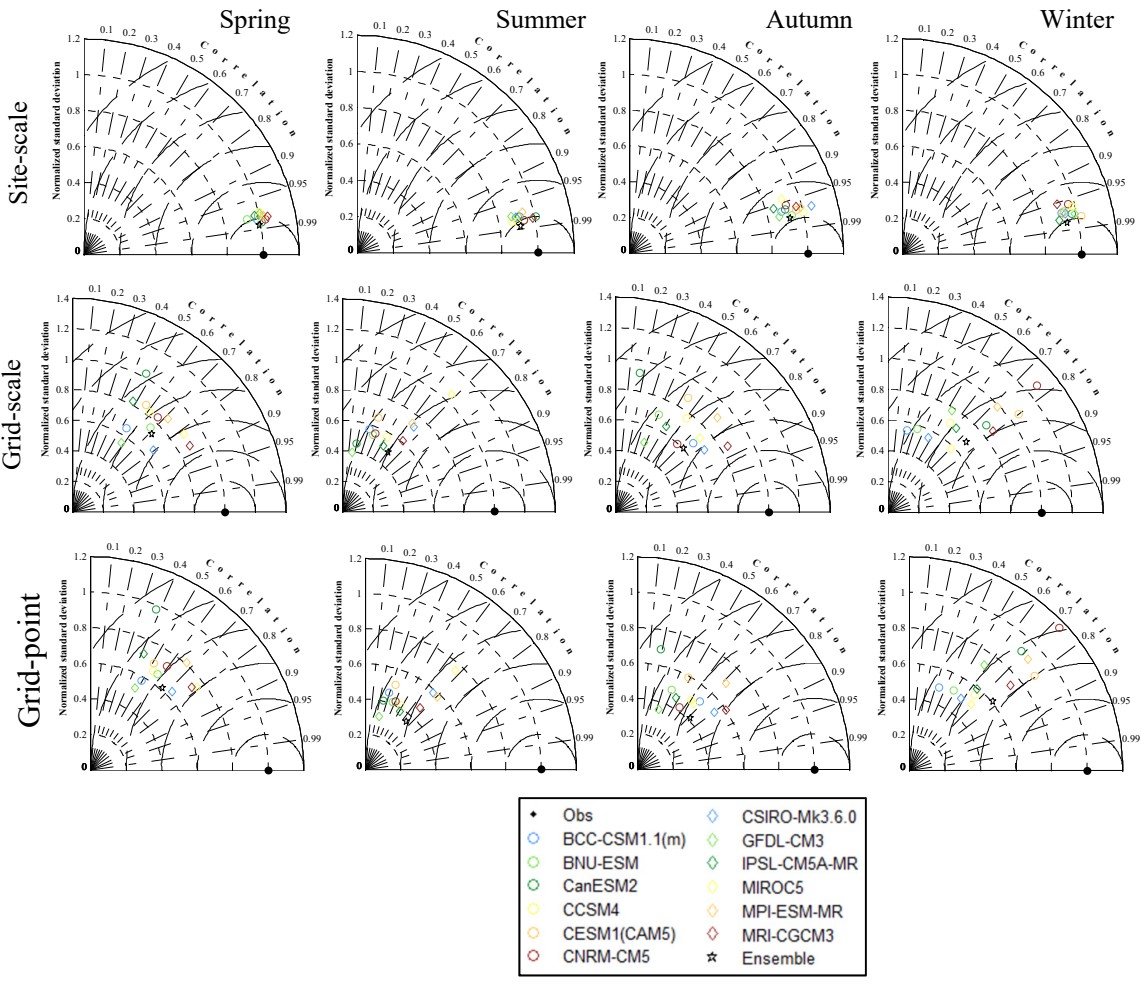

5 **Figure 6. Taylor diagrams of seasonal SDII derived from GCMs against observations based on the site-scale, grid-scale, and grid-point comparisons.**





**Figure 7. Box-and-whisker plots of areal mean of CDD (day), NW (day), R5d (mm), and SDII (mm/day) across China of observations, historical, RCP2.6 and RCP8.5 scenarios at the site-scale (the first column), grid-scale (the second column), and grid-point comparisons (the third column). Precipitation extremes of observations and historical scenario are during the period of 1991-2005, and those of RCP2.6 and RCP8.5 scenarios are during the period of**
10 **2071-2100.**







**Figure 8.** Spatial distribution of difference of simulated precipitation extremes relative to observations, i.e. $p_{GCM} - p_{obs}$, for CDD (day), R5d (mm), and SDII (mm/day) during 1991-2005 at the site-scale and grid-point comparison.











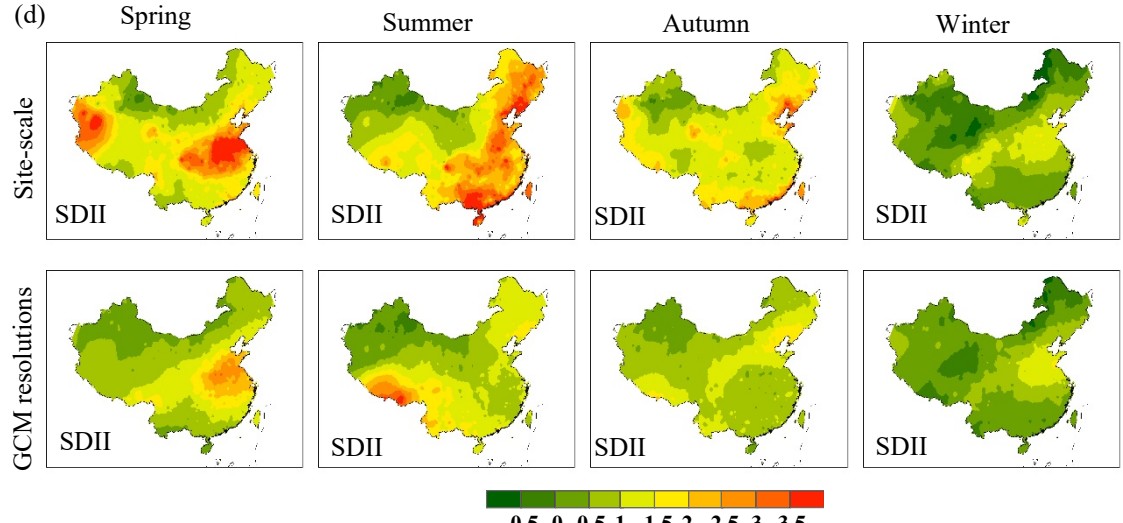

**Figure 9. Projected changes in (a) CDD (day), (b) NW (day), (c) R5d (mm), and (d) SDII (mm/day) during 2071-2100 under RCP8.5 relative to during 1971-2000 under the historical scenario at the site scale and at GCMs' original spatial resolutions.**