# Peer review of "Impacts of spatial resolutions on projected changes in precipitation extremes: from site- to grid- scales"

_Hydrology and Earth System Sciences, 2017_

## Referee Comment (RC1) · Anonymous Referee #1 · 3 Aug 2017

This study tried to verify the resolutions of GCMs have less effect on projected changes in precipitation extremes. I thought the current conclusion did not make sense due to GCMs with coarse resolutions showed bad performance in simulating the extremes. One approach was downscaling tech including dynamic and statistic methods. The statistic was not appropriate in western China because lack of observations. So this study still focused on the GCMs' resolution impact on climate projection was not a new insight, even combined with statistic downscaling. Here, the statistic downscaling could NOT improve the GCMs resolution too much because of the observations were very limited derived from 509 stations. Some studies have already verified the finer resolution through dynamic downscaling could improve the model performance in terms of

precipitation and the other elements, especially captured the extremes event induced by mesoscale system. And their results also showed different resolutions had great impact to modeling climate change (e.g. Gao et al. 2012 Climate Research). So I suggest the contributors to turn to the GCM-RCM through dynamics downscaling and see if the spatial resolutions have impact on projected changes.

—————————————————————

---

## Short Comment (SC1) · 3 Aug 2017

1) From the reader's point of view, in the current version of the paper, the authors' thoughts are scattered. The introduction of the paper needs to be re-structured. The paragraphs are not well inked to convey the need of the research. For example, the first paragraph of the introduction is ended with the C-C relationship. However, the ending of the first paragraph does not lead to the content of the second paragraph.

2) With warmer temperature, the saturated vapor pressure is increased. Consequently, the air will have more capacity to hold more water vapor. However, from the reader's point of view, although the water vapor is the main striking factor that determines the

formation of precipitation, it is not the only factor that defines the formation of precipitation. In other words, there are many vital factors that need to be considered. Therefore, considering the slope (see page-7) as the defining criteria may not comprehensively unveil the outcome of the research.

3) As per the authors, changes in extreme climate can lead to significant impacts on the occurrence and severity of natural disasters which will result in changes in risk of failure for urban infrastructures (see line 1-2 in page-2). However, the introduction of the manuscript does not clearly outline the reason(s) for considering the precipitation extreme as the extreme climate. From the reader's point of view, precipitation (or precipitation extreme) is a subset of many interrelated factors that determine the extreme climate.

4) From the reader's point of view, the last paragraph does not fit the section (i.e., introduction).

5) From the reader's point of view, in the current version of the paper, some of the terminologies (e.g., extreme precipitation) are not well defined. What is meant by extreme precipitation? Are the authors referring to the indices (see Table 2)?

6) From the reader's point of view, some of the methodologies adopted need to be explained in detail. Without knowing the exact methodology(s), evaluating the outcome of the research may not be fruitful. Therefore, simple example(s) to illustrate some of the methodologies is expected to enhance the readability.

7) As per the authors, although most previous studies agreed on projected increases in future precipitation extremes, they hardly made agreements on the changing rates because they were based on either different GCMs or resolutions (see line 13-14 in page-2). From the reader's point of view, the authors' statement is not well understood. What is meant by changing rates? What is meant by they hardly made agreements? What is the current status of the literature? Has this topic (i.e., changing rates) already been researched by other researchers?

8) From the reader's point of view, the title of the manuscript does not reflect the content (see line 25-29 in page-3) of the manuscript.

9) As per the authors, datasets of precipitation extremes at various spatial resolutions from the site scale to the 4o×4o gridded scale are constructed by aggregating daily precipitation of stations (see line 25-26 in page-5). What is the site scale? Why did not the authors use one of the existing interpolation methods (see line 6 in page-4)?

10) What is the unit of SDII (see line 26 in page-6)?

———————————————

---

## Author Comment (AC1) · 14 Aug 2017

We appreciate the reviewer's comments and suggestions. We think the points raised by the reviewer have been well known and discussed in others' and our previous studies. Our analysis of the current study also has well considered these points and our conclusions do not contradict the reviewer's concerns. More importantly, to the best of our understanding on his/her comments and reference, we think the questions of our manuscript tries to answer are different from the reviewer's concerns in nature. Below are our point-by-point responses to clarify how our analysis supports the reviewer's comments and how our objectives are different from the reviewer's concerns.

[Figure]

(1) Reviewer: I thought the current conclusion did not make sense due to GCMs with coarse resolutions showed bad performance in simulating the extremes.

Reply: Our comparisons of extremes estimated from GCMs at grid resolutions of 2.5ox2.5o against observation shown in Figs. 4-7 indicate that GCMs with coarse resolutions are not very good at representing extremes, showing that we have fully considered this factor. In fact, our conclusion on insensitivity of pfut-phis (absolute value of an extreme in a future period minus that in a historical period) to spatial resolutions are developed based on the analysis on observational data (as shown in Section 4.1), which was often used as a reference in previous studies. In that part, we interpolated the daily precipitation of station-based observations into precipitation of grid cells with various spatial resolutions and hence estimate pfut-phis based on daily precipitation with various spatial resolutions. We found that pfut-phis estimated at different resolutions change marginally. The design of this experiment was based on the procedure used by Chen and Knutson (2007) published in Journal of Climate, in which sensitivity of p (absolute value of an extreme) to spatial resolutions was discussed. In other words, the validity of the conclusion does not rely on GCM performances. Therefore, it will be appreciated if the reviewer provides more details on the statement "I thought the current conclusion did not make sense due to GCMs with coarse resolutions showed bad performance in simulating the extremes", so as we can better improve our manuscript.

(2) Reviewer: The statistic was not appropriate in western China because lack of observations.

Reply: We explicitly indicated the lack of observations in western China in Line 3-5 Page 4 in the manuscript, showing that we have considered this factor in the observational data. In the additional analysis of this response (point 7), the impacts of the varying density of stations in different regions on our analysis are further discussed based on the updated dataset with 2,088 stations. In our analysis, the grid cells with various resolutions are constructed centered on each of the stations. The numbers of stations covered by grid cells at various resolutions are shown in Fig. R1b below for

the updated 2,088 stations. The mean of the numbers of stations are around 20 and 40 for 2ox2o and 4ox4o resolutions, respectively, indicating that the numbers of stations are sufficient to estimate the values of grid cells. At the same time, the range of the 10th and 90th percentile of the numbers stations reflects the influence of densities of stations in different areas. This paper aims to generalize the aggregated conditions of sensitivity to spatial resolutions over a vast area, rather than to estimate extremes in a specific region. Therefore, the high density of stations in eastern China can balance the impacts of low density in western China. Fig. R1b will be added in the revised manuscript for readers to better understand the uncertainties.

(3) Reviewer: So this study still focused on the GCMs' resolution impact on climate projection was not a new insight, even combined with statistic downscaling.

Reply: We need to clarify that the focus of the study is impacts of spatial resolutions in general (NOT just GCMs, but also different types of datasets, such as reanalysis and observations, at different resolutions) on changes in extremes (here we only focus on pfut-phis.) based on spatial interpolation. In fact, we mainly used observational data to derive our conclusions (see our reply to point 1). Furthermore, as discussed in our introduction in L25 P2 – P30 L3, spatial resolution impact on climate projection is a popular topic and at the same time very controversial, because many scientific and technical details are involved, for example, dynamical downscaling or statistical downscaling (e.g. Fowler et al., 2007), usages of different RCMs or different statistics for downscaling (e.g. Maraun et al., 2010), orders of interpolating daily precipitation from original resolutions first then estimating extremes vs estimating extremes at original resolutions first then interpolating the extremes (e.g. Chen and Knutson, 2007), comparison of extremes at different resolutions vs at an uniform resolution (e.g. Hu et al., 2016), etc. Different choices of these techniques may lead to very different results. Therefore, although this issue is not new, it still needs more studies to improve our understanding.

This study focuses on the problems in comparing extremes estimated from daily precipitation at different spatial resolutions, rather than simply studying statistical down-scaling. We estimate changes in extremes derived from daily precipitation of different datasets, e.g. GCMs, interpolated datasets, etc., at their raw resolutions, and then quantify to what extent changes in extremes at different resolutions are comparable. This study aims to address the gap of the previous studies which have shown sensitivity of magnitudes of extremes (i.e. p) derived from daily precipitation at various resolutions. Here we (a) quantify the sensitivity of pfut-phis derived from daily precipitation in datasets at various spatial resolutions and estimate (b) how this factor affects the comparison of extremes estimated from datasets at different resolutions. As we know, a large number of previous studies compared extremes at various resolutions based on statistical downscaling, interpolation, or direct comparison at various resolutions (e.g. Maraun et al., 2010; Hu et al., 2016). To the best of our knowledge on different types of solutions on comparing extremes at various resolutions, the scientific community does not reach consensus on which solution is better than the others. We expect there will be more studies on changes in extremes based on different methods. Therefore, this study is urgently needed and has certain scientific and practical merits by improving our understanding in impacts of resolutions on estimating changes in extremes.

(4) Reviewer: Here, the statistic downscaling could NOT improve the GCMs resolution too much because of the observations were very limited derived from 509 stations.

Reply: To address this concern, we have updated our dataset to include daily precipitation of 2,088 stations across China with good quality control in this response. We conduct preliminary analysis based on these 2,088 and find out that the conclusions are still the same as those derived from the 509 stations (see additional analysis in point 7). In the future revised manuscript, we will update all results with the 2,088 stations. We think these 2,088 stations can address the reviewer's concern.

(5) Reviewer: Some studies have already verified the finer resolution through dynamic downscaling could improve the model performance in terms of precipitation and the

other elements, especially captured the extremes event induced by mesoscale system. And their results also showed different resolutions had great impact to modeling climate change (e.g. Gao et al. 2012 Climate Research).

Reply: We totally agree that models implemented and run in a finer resolution can better represent the precipitation and extremes, because models implemented and parameterized in a finer resolution can better describe the local and regional characteristics and processes, such as the parameteriazation of mirco-physics, clouds formation, and land surface.

However, the question the study tries to discuss is to assess the relative changes in pfut-phis ESTIMATED from daily precipitation at various spatial resolutions based on spatial interpolation and statistical downscaling. In other words, we already have a number of datasets at different spatial resolutions (although we only consider interpolated observations and GCMs, these can be GCM outputs, reanalysis datasets, interpolated products, remote sensing products, etc. in a more general perspective), and then changes in extremes are estimated based on daily precipitation of these datasets. Afterwards, we discuss how differences are pfut-phis estimated at various spatial resolutions. The question is important to understand how comparable are changes in extremes estimated by daily precipitation interpolated to different resolutions for different purposes, given that a large number of previous studies estimating extremes based on interpolation and statistical downscaling. The design of the experiments follows Chen and Knutson (2007) published in Journal of Climate, in which the impacts of spatial resolutions on estimation of the absolute values of extremes were examined. The conclusions of our study agree with the findings of Chen and Knutson (2007) that spatial resolutions have considerable impacts on estimating the absolute values in extremes (i.e. p), e.g. extremes are found to be smaller estimated at coarser resolutions. Building up on this understanding, our conclusions further suggest that the changes of extremes between two periods, i.e. pfut-phis (absolute value of an extreme in a future period minus that in a historical period), have relatively less sensitivity to spatial

resolutions compared to the absolute values.

(6) Reviewer: So I suggest the contributors to turn to the GCM-RCM through dynamics downscaling and see if the spatial resolutions have impact on projected changes.

Reply: As far as we know, we hardly see any consensus of scientific community that dynamical downscaling is better than statistical downscaling and vice versus. In fact, as shown in our discussion, these methods have their own advantages and disadvantages (L10-L30 P3). We agree that the impacts of spatial resolutions based on GCM-RCM dynamical downscaling is a very important topic, but our study focuses on impacts of resolutions on changes in extremes by interpolating a larger number of observations into different resolutions. Given many previous studies were based on interpolation and statistical downscaling, we believe the angle and way that we analyze the sensitivity of pfut-phis to spatial resolutions is necessary to improve our understanding of impacts of resolutions on changes in extremes.

(7) Authors: Additional analysis based on the 2,088 stations across China

To address the concern on the number of stations of observed daily precipitation, we here in this response have collected daily precipitation of more than 2,000 stations of 1961-2005 across China. After quality control, 2,088 stations are selected as shown in Fig. R1a. Similar to the 509 dataset we used previously, the stations are located more in the eastern China and less in the western China. Followed the procedure introduced in the manuscript (L25 P5 – L5 P6), datasets of precipitation extremes at various spatial resolutions from the site scale to the 4ox4o gridded scale are constructed by aggregating daily precipitation of stations covered by grid cells centered at each of the meteorological stations. The relative changes of extremes between 1986-2005 and 1961-1985, i.e. p1986-2005-p1961-1985, are estimated from grid cells at various resolutions. The averaged numbers of stations covered by grid cells are about 20 and 60 at 2ox2o and 4ox4o resolutions, respectively (Fig. R1b). The 10th and 90th percentile at 2ox2o the resolution are 5 to 40, respectively, which are due to the spatially varying

Interactive
comment

density of stations in different parts of China. The averaged p1986-2005-p1961-1985 and p1961-1985 across stations from the 2,088 and 509 datasets are compared in Fig. R2 (only CDD and SDII are shown in this response). It is observed that the changes in p1986-2005-p1961-1985 estimated by grid cells at various resolutions are much smaller compared to those in the magnitudes of extremes, i.e. p1961-1985, in both datasets of 2,088 and 509 stations. To better illustrate the sensitivity analysis, an example is shown in Fig. R3. Fig. R3a shows the grid cells with 2ox2o and 4ox4o resolutions centered at a randomly selected station. The number of stations covered by the constructed grid cell increases as the size of the grid cell increases (Fig. R3b). In the typical 2ox2o resolutions of GCMs, about 60 stations have been covered by the 2ox2o grid cell. These 60 stations are aggregated to generate the CDD1961-1985 and CDD1986-2005-CDD1961-1985 of the 2ox2o grid cell. The procedure is repeated to estimate extremes with spatial resolutions from the site to 4ox4o, and the values of various resolutions are plotted in Fig. 3c. Again, in this example, we can observe that the magnitude of CDD decreases considerably from 62 days to 53 days as grid cell size increases to 4ox4o, while the change of CDD fluctuates from -3 to -6 days in these grid cell sizes. Therefore, the changes in extremes are less sensitive to spatial resolutions compared to the magnitudes.

We will complete a more comprehensive analysis based on the 2,088 stations in the revised manuscript. Furthermore, we will also incorporate the above responses into the revised manuscript to better describe our objectives and the differences of our study from previous studies.

Key references:

Fowler, H.J., Blenkinsop S. and Tebaldi C.: Review: Linking climate change modelling to impacts studies: recent advances in downscaling techniques for hydrological modelling. Int. J. Climatol., 27, 1547-1578, 2007.

Hu, Z., Hu Q., Zhang C., Chen X., and Li Q.: Evolution of reanalysis, spatially interpolated and satellite remotely sensed precipitation data sets in central Asia. J. Geophys. Res. Atmos., 121, 5648-5663, 2016.

Maraun, D. et al.: Precipitation downscaling under climate change: recent developments to bridge the gap between dynamical models and the end user. Rev. Geophys. 48, RG3003, doi:10.1029/2009RG000314, 2010.
* * *
[Figure]

Figure R1. Locations of meteorological stations across China (a) and number of stations covered by grid cells with various resolutions centered at each of the stations (b). In (b), the solid line is the averaged number of stations coverved by grid cells centered at the 2,088 stations, the dashed lines are the 10th-90th percentile.

[Figure]

Figure R2. Average magnitudes of 1961-2005 $p_{1961-1985}$ (orange) and changes between 1986-2005 and 1961-1985, i.e. $p_{1986-2005} - p_{1961-1985}$ (blue) of CDD (day) and SDII (mm/day) derived from the 2,088 and 509 stations at various spatial resolutions in degree.

[Figure]

Figure R3. An example of grid cells construction and sensitivity of $CDD_{1961-1985}$ and $CDD_{1986-2005}$-$CDD_{1961-1985}$ in a station. (a) Grid cells with spatial resolutions of 2°x2° (the red inner domain) and 4°x4° (the red outer domain) centered at the station (the red point). (b) Number of stations covered by grid cells with various resolutions centered at the station. (c) $CDD_{1961-1985}$ and $CDD_{1986-2005}$-$CDD_{1961-1985}$ estimated by values of grid cells at various resolutions centered at the station.

---

## Referee Comment (RC2) · Anonymous Referee #2 · 14 Oct 2017

The manuscript described the potential impact of model spatial resolution on the projected future changes in precipitation extremes using daily rainfall observation and simulation over China. The authors argued that even though the precipitation related extreme indices are sensitive to spatial resolution of analyzed data, the impact on future projection is relatively small. The authors also used three different approaches to compared and evaluated the model gridded mean output with mismatched point observation. The result suggests that by applying statistical downscaling method to model simulation to derived the precipitation extremes to the observed location outperformed the direct comparison of model simulated gridded output and station observation with and without scaling the station and model data to a common grid size. They also

highlighted the similarity between the simulation and observation and implied that the difference in the data spatial resolution does not matter in the comparison.

While there is certainly some interests on the topic the authors tried to address, the presentation and discussions in the paper are either already known in previous literature or missed the real important issues in such study. In particular, the authors totally neglect the potential mixed influence from the model bias (or spread) and model spatial resolution. The necessity to exclude the spatial mismatch of the data is the basis for fair comparison. One should not argue if in practice the result are similar (due to various reasons), then one don't have to make the comparison in right way. The more detailed comments are listed below. I can't really find anything new reported in the paper. Further, the discussions are often misleading. Therefore, I would recommend the rejection of paper.

Originality: Fair Technical quality: Fair Clarity of presentation: Fair Significance: Poor

General comments:

1. The impact of spatial resolution on rainfall extremes (or even more detailed rainfall intensity–duration–frequency relationship) from point measurement to large area average are well known in the previous hydrological study. Due to such impact, one should only compared (or validate) the rainfall extremes at the same resolution. Therefore, the three approaches used by the authors can only be considered as how incorrect the comparison can be, especially the model simulations run at various resolutions and different from point measurement from station. Nevertheless, the authors tends to emphasize the relatively small impact from such spatial scale differences for both model evaluation and future projection. But that is very misleading in term of basic principle.

2. The most important issue related to the mismatched spatial resolution of daily data in calculating precipitation-related extreme indices from CMIP5 models is whether the spread of model projected future change is truly due to model difference, not the model resolution. This is often overlooked. For example, even the papers cited in the IPCC
AR5 report regarding the projected future change of rainfall extremes (Skillmann et al. 2013a,b, JGR Atmos., DOI: 10.1002/jgrd.50203, DOI:10.1002/jgrd.50188) did not take into account the model resolution difference when all of climate extreme indices are calculated from model daily output at original model resolution. Indeed, the range of model projected change might not be that different from the result based on upscaled or downscaled data due to mixed impact from model bias and model spatial resolution. However, one should carefully clarify such mixed impact into detailed. In that regard, the authors did not provide any useful insight to the issue.

3. There is no surprise for the authors to find that using statistical downscaling method to transfer the model output to the station location. perform better since the quantile mapping procedure correct the model bias. What is the point for the authors to compare with other approaches that with only simple interpolation and upscaling or even without upscaling the station measurements.

4. In addition, the authors often show only model ensemble mean projected change, but as point out earlier the model ensemble mean might be affected differently by model tuning at higher and lower spatial resolution. Further, it is also expected to have minimum impacts from different approaches on the future projected percentage change since the impact largely canceled out when the same operators are applied to both denominator and numerator.